# Inhibition of Wnt signaling in primary human hepatocytes promotes *Plasmodium falciparum* liver stage development

Abhishek Kanyal[1☉], Geert-Jan van Gemert[2☉], Haoyu Wu[1,3☉], Alex van der Starre[2], Johannes H.W. de Wilt[4], Teun Bousema[2], Robert W. Sauerwein[2], Richárd Bártfai[1☉*], Annie S. P. Yang [2☉¤*]

**1** Department of Molecular Biology, Faculty of Science, Radboud Institute for Molecular Life Science, Radboud University, Nijmegen, The Netherlands, **2** Radboud Center of Infectious Diseases, Department of Medical Microbiology, Radboud University Medical Center, Nijmegen, The Netherlands, **3** CAS Key Laboratory of Regenerative Biology, Guangzhou Institutes of Biomedicine and Health, Chinese Academy of Sciences, Guangzhou, China, **4** Department of Surgery, Radboud University Medical Center, Nijmegen, The Netherlands

☉ These authors contributed equally on this work.
¤ Current address: Leiden University Center for Infectious Diseases (LUCID), Leiden University Medical Center, Leiden, the Netherlands
* r.bartfai@science.ru.nl (RB); a.s.yang@lumc.nl (ASPY)

## Abstract

After infection of the human host, the first stage of the *Plasmodium falciparum* (Pf) lifecycle takes place in the liver. Understanding of host-parasite interactions during liver stage development is compromised by the rapid loss of functionality and Pf permissiveness of cultured primary human hepatocytes (PHHs). Here, we substantially delay the loss of Pf permissiveness by using a medium containing serum-replacement and signal transduction inhibitors. We analyzed and integrated transcriptomic profiles of cultured PHHs with the phenotypic presentation of developing Pf liver stages, revealing a number of host signaling pathways that contributed to dedifferentiation of hepatocytes and influenced Pf liver stage development. In particular, inhibition of the Wnt pathway showed a significant positive impact on size and maturity of Pf liver stage schizonts, while retaining the metabolic activity and epithelial nature of PHHs. Therefore, our study provides insights into hepatocyte characteristics that are important for Pf permissiveness and an improved *in vitro* liver stage model. This should facilitate identification and development of novel therapeutic strategies for Pf liver stages.

### Authors' summary

Despite substantial elimination and control measures, malaria still causes mortality and global disease burden. Liver stage development of the malaria parasite is a critical first step preceding clinical symptoms and is a potent target of

**Data availability statement:** The next-generation sequencing dataset associated with this study has been uploaded to Gene Expression Omnibus with the publicly available accession series GSE263643. The dataset comprises of: The metadata sheet with information on experiments and samples, Raw data files: fastq files for RNA-sequencing, Processed data files: i) raw counts files generated from Combat-seq and ii) normalized count files generated from DESeq2.

**Funding:** The salary of A.S.P.Y was supported by the Dutch Research Council (Nederlandse Organisatie voor Wetenschappelijk Onderzoek) talent program (VI.Veni.192.171) and ZonMW Off-Road grant (#04510012010050). A.K. was supported by the Dutch Research Council (ZonMW-TOP-grant #91218010, recipient RB). Work in the Bártfai lab has been supported by an ERC Synergy grant (#101118536). The funders had no role in study design, data collection, analysis, decision to publish or preparation of the manuscript.

**Competing interests:** The authors have declared that no competing interests exist.

antimalarial vaccine/drug development. However, we lack sufficient understanding of critical components of the host hepatocyte environment supporting or preventing liver stage parasite development. In this study, we observed substantial decline in the ability of cultured primary human hepatocytes to support parasite development and identified molecular signatures and altered activity of host signaling pathways associated with this loss of permissiveness. Specifically, we demonstrate the impact of the Wnt pathway on the growth and maturation of liver stage parasites in cultured primary human hepatocytes. We thus provide insight into how host hepatocyte homeostasis can impact liver stage parasite development and pave the way for further dissection of host-pathogen interactions and development of improved intervention strategies.

## Introduction

The mosquito-borne disease, malaria, remains a devastating global burden with approximately 250 million cases and 620,000 deaths annually [1]. Most of the deaths are due to the parasite *Plasmodium falciparum* (Pf), which establishes itself inside hepatocytes as a liver stage after the initial deposition of sporozoites into the host skin by an infected mosquito. Over a period of approximately one week, the intracellular liver stage parasite replicates and differentiates into a mature schizont, that contains approximately 90,000 exoerythrocytic merozoites capable of invading red blood cells (RBCs) [2]. The release of these merozoites into the circulation and their subsequent cyclical asexual replication in RBCs is responsible for all clinical symptoms associated with the disease.

Despite its fundamental importance in enabling effective infection, our understanding of the complex parasite-host interactions within the infected hepatocyte remain elusive. Studies with the rodent malaria model, *Plasmodium berghei* (Pb) model have been informative as a proxy for Pf. Amongst others, specific host membrane receptors have been identified to be involved in Pb sporozoite invasion of hepatocytes such as CD81 and Scavenger Receptor B1 (SR-B1) [3–5]; similarly, host factors involved in nutrient acquisition [6–9] and/or prevention of host cell death pathways [10–13]. Yet, findings in Pb are often not translatable to Pf-host interactions: the rodent parasite has only a 2-day liver stage maturation time compared to 7-days for Pf [14].

Research in Pf liver stage development primarily involves *in vivo* models such as mice with humanised liver [2,15,16], or *in vitro* models where hepatocyte cell-lines such as HC-04 [17,18] and primary human hepatocytes (PHHs) are used. The use of cell lines has been limited to the identification of potential host receptors involved in parasite invasion such as EphA2 [3,19] and glypican 3 [18], but the exact invasion route remains to be unclear. PHHs are considered the gold standard in drug assays [14,20] and were used to identify host factors needed for the development of Pf liver stage schizonts such as SR-B1 [9,21,22] and glutamine synthetase [23]. A well-known limitation of *in vitro* cultured PHHs, however, is the quick loss of hepatic features [24,25], yet

the impact of this has not been explored in Pf liver stage development and may hinder the discovery of host factors that are important for full maturation of liver stage schizonts. Establishment of an *in vitro* hepatocyte model that preserves hepatocyte features is therefore needed. While the loss of hepatic features is largely due to the deregulation of specific host signaling pathways [24], their potential impact on parasite development have not been previously examined.

Here, we set out to 1) identify the impact of loss of hepatic functions in *in vitro* PHHs and their permissiveness to Pf infection, 2) improve and stabilize hepatocyte culture conditions for Pf permissiveness, facilitating complete schizont development and 3) identify host pathways in hepatocytes that influence the developmental kinetics of Pf liver stages. Using RNA sequencing to compare the transcriptomic profiles between freshly isolated and cultured PHH, we identified upregulation of the Wnt (Wingless and Int 1), TGF-β (Transforming Growth Factor Beta), Notch and Rho signaling pathways. Treatment with specific chemical antagonists of some pathways led to an improvement in Pf schizont development. Furthermore, the Wnt pathway was shown to be most influential in determining the size and maturation of the Pf liver stage schizont.

## Results

### *In vitro* cultured PHHs lose permissibility *for* Pf liver stage infection and development

Traditionally fresh PHHs are cultured in standard medium containing human sera (William's B with Human sera; WBH) [23,26,27] and infected with Pf 48 hours after plating by addition of sporozoites. However, PHHs quickly lose their cellular identity, and this may impact permissibility to Pf infection. To investigate the effect of culturing time on Pf permissiveness, we infected PHHs at different time points post-plating (p.p.) with Pf strains, chosen for their differences in developmental kinetics (PfNF54, PfNF175 and PfNF135 [23]). Schizont numbers were evaluated at five days post-infection (p.i.) (Fig 1A). We observed a sharp decrease in number of schizonts in relation to the p.p. period for all Pf strains tested (n = 2 for PfNF54 and PfNF135; n = 1 for PfNF175) with the most drastic decline occurring between days 2 and 5 p.p. This reduction could not be explained by a decrease in the number of viable hepatocytes, which became prominent only after day 9 p.p. (Fig 1C).

A previous study showed that medium containing human serum negatively affects hepatocyte functionality [21]. Therefore, we examined the impact of serum on hepatocyte permissiveness for parasite liver stage development via the removal of sera (WB versus WBH). Mild non-significant improvements in schizont numbers and the presence of the maturation marker PfMSP1 were observed in serum negative medium (WB; S1 Fig). We hence concluded that a novel hepatic culture medium is needed to restore schizont numbers to those seen on day 2 p.p. A serum replacement supplement (B27) on a William's E Glutamax base was tested given its beneficial effect on the differentiation of stem cells [28] and hepatic organoids into hepatocytes [29,30] as well as other hepatic culture systems [24]. Addition of B27 to the basal medium William's E Glutamax substantially delayed the loss of hepatocyte permissiveness by at least 7 days (Fig 1D). This was not due to changes in hepatocyte numbers, which remained comparable to the WBH medium thus reflecting a change in the phenotype of the PHHs (Fig 1E).

To understand the cellular changes in PHHs that may influence Pf permissiveness and intracellular development, we compared the transcriptional profile of freshly isolated PHHs (n = 2) with uninfected PHHs cultured in B27 for 7 days p.p.. This time point was selected since at this point the PHHs display a clear reduction to Pf permissiveness even in B27 media. The transcriptional profile showed an upregulation of 2403 gene transcripts and a down regulation of 2732 genes (Fig 1F; cultured versus fresh). The most significantly up-regulated genes were those involved in actin filament organization, cell-substrate adhesions, wound healing and extracellular matrix organization (Fig 1G). All these pathways are classically associated with the epithelial to mesenchymal transition (EMT), which is a sign of trans-differentiation [3]. As for down-regulated genes, we found a strong enrichment for those involved in metabolism, i.e., fatty acid metabolism, lipid catabolism, organic anion transport and alcohol metabolism, among others (Fig 1H). These pathways are linked with metabolic functionalities of hepatocytes, and their downregulation indicates a decay of core hepatic traits. This could additionally alter the amount and type of metabolites available for the liver stage parasite. Together, while serum replacement with B27 showed a substantial positive effect on PHH permissiveness to Pf infection, the transcriptional profile obtained indicates significant differences from freshly isolated PHHs, pointing to a loss of hepatic functions with potential effects on size and maturation of liver schizonts.

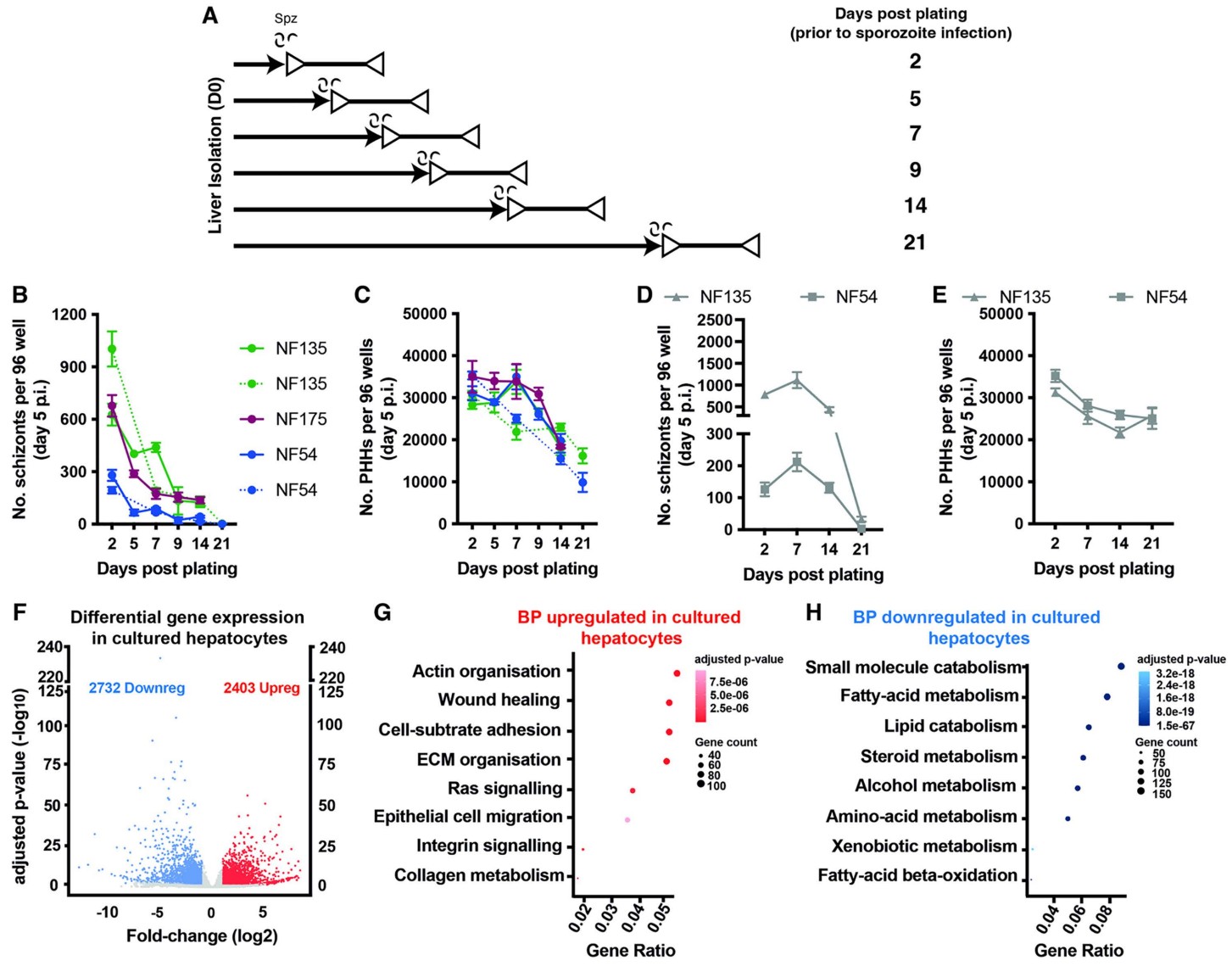

**Fig 1. Pf-schizont development post-hepatocyte plating and transcriptomic profiles of primary human hepatocytes (PHH). A)** The solid arrows show the number of days after liver isolation (post-plating) and the lines with the clear triangles indicate the period of schizont development. **B)** The number of schizonts in PHH on day 5 post infection (p.i.) when infected on different days p.p. for parasite strains: NF175 (magenta;), NF135 (green;) and NF54 (blue) in two independent experiments (except for NF175). **C)** The number of PHHs present on day 5 p.i. on indicated days p.p for the specific parasite strains. For each independent experiment, the mean of three replicates is shown along with the standard deviation. The number of schizonts (D) and hepatocytes (E) on day 5 post invasion (p.i.) for NF135 and NF54 (n = 1) for different infection times post plating (p.p.) cultured in B27 medium (grey). **(F)** Volcano plot highlighting the differentially expressed genes between hepatocytes cultured in B27 medium ("mock") and freshly isolated hepatocytes ("fresh"). Dot-plots depicting the biological processes enriched amongst upregulated (G) and downregulated (H) genes in cultured hepatocytes as compared to freshly-isolated hepatocytes.

## Specific host signal transduction pathways influence Pf liver stage development

To narrow down the regulatory processes that could lead to the observed gene expression changes we filtered our gene ontology enrichment analysis (for genes upregulated in cultured PHH) for hits against signaling pathway terms. Genes upregulated in cultured PHHs (Fig 2A) showed a strong enrichment for signaling pathways via Wnt (Wingless and Int-1),

Ras, transforming growth factor beta (TGF-β), apoptosis and Notch. These pathways were chosen for further investigation due to the availability of commercial agonists and antagonists, their published role in hepatocyte functionalities and the significance of their upregulation [24]. Compared to the fresh PHH, genes involved in the Wnt, TGF-β and Notch pathways were strongly upregulated (Fig 2B-D) in cultured PHH, while Bone Morphogenic Protein (BMP) and adenylyl-cyclase pathways were mildly upregulated in B27 cultured PHH (Fig 2E). While some donor-to-donor variation in the deregulation of individual genes was observed, a general up-regulation for all pathways was apparent (Fig 2B-E). To examine the impact of these host pathways on Pf liver stage development, we treated PHH for 6–8 days p.p., with selective agonists and antagonists of the individual pathways. Wnt signaling was inhibited with IWP2, TGFβ signaling was inhibited with SB431542, Notch signaling was inhibited with DAPT, BMP signaling was inhibited with LDN193189 and adenylyl-cyclase signaling was activated with Forskolin (Fig 2F-2J). Quantitative measurements assessed the number (Fig 2F), the replication (size, nuclear DNA content via DAPI; Fig 2G and 2H), health (human glutamine synthetase: hGS; S2B-C Fig) and maturation (PfMSP1; Fig 2I) status of the liver stage schizonts, respectively. Modulation of both the BMP and the adenylyl-cyclase pathways through the usage of LDN193189 and Forskolin, respectively did not result in significant improvements in any of the parameters tested. DAPT-treated hepatocytes showed significant improvements in the number and size of schizonts observed but not in other parasite phenotypic parameters. Improvements were seen for SB431542 treatment for all measured parameters most notably in the number of schizonts and expression of the maturation marker, PfMSP1. Similarly, IWP2-treated hepatocytes showed significant improvements in the schizont size, DNA content per schizont and expression of PfMSP1 per schizont but no significant increase in schizont numbers. Finally, we tested a previously published cocktail containing all five of these inhibitors (5C) [24] (S3 fig). Hepatocyte monolayers cultured in the presence of 5C showed poor support of PfNF175 schizont size and numbers (S4D-J Fig) contrary to its beneficial effect on viral infection reported earlier [24]. Together, these data demonstrated that inhibition of specific host pathways, and particularly Wnt (inhibited by IWP2), TGF-β(inhibited by SB431542) signaling are important for supporting Pf liver stage permissiveness and/or schizont development in PHHs. Here, we chose to further focus on the effect of the Wnt pathway as it had a more profound effect on parasite size and maturation.

### Inhibition of the Wnt signaling pathway helps preserving the metabolic profile and epithelial nature of PHHs and improves the developmental kinetics of Pf liver stage development

To study the earliest impact of Wnt pathway suppression on Pf liver stage development, PHHs were treated with IWP2 6–8 days prior to and during Pf infection by three different parasite strains (Fig 3A-3F). IWP2 treatment promoted parasite development (size, nuclear content and PfMSP1 expression), from day 5 p.i. onwards for all strains tested. To further capture the transcriptional changes associated with IWP2 treatment, RNA sequencing was performed on hepatocytes cultured in B27 ("mock") or treated with IWP2 at day 5 p.i. as well as on freshly isolated PHHs as a reference point. Furthermore, we also analysed the transcriptional changes brought about by 5C to understand why treatment with this cocktail was not beneficial for parasite growth. The original experiment was performed with five replicates per culture/treatment condition, but upon closer inspection we excluded replicate 2 from our analysis owing to suboptimal quality (S6A Fig). Based on spatial proximity in the PCA space, B27 cultured cells were very distinct from freshly isolated hepatocytes (67% of variance in the PC1 axis, Fig 3G) as also evident from Fig 1F–1H. Upon IWP2 treatment we observed a notable shift of B27 cultured samples towards fresh samples on the PCA space although it remained closer to B27 than to fresh (Fig 3G). To quantify this shift, we calculated the sample distance scores which indicated an increased similarity between IWP2-treated cells and fresh hepatocyte (mean distance score: 147) as compared to B27 cultured cells (mean distance score: 168) (lower scores define higher similarity between samples). 5C treatment elicited additional transcriptional changes as indicated by an increased separation of the samples away from fresh hepatocytes along the PC2 axis (Fig 3G). In IWP2-treated cells, 394 genes were upregulated, and 344 genes downregulated relative to B27 cultured cells, (Fig 3H). Gene Ontology analysis of the genes

PLOS Pathogens

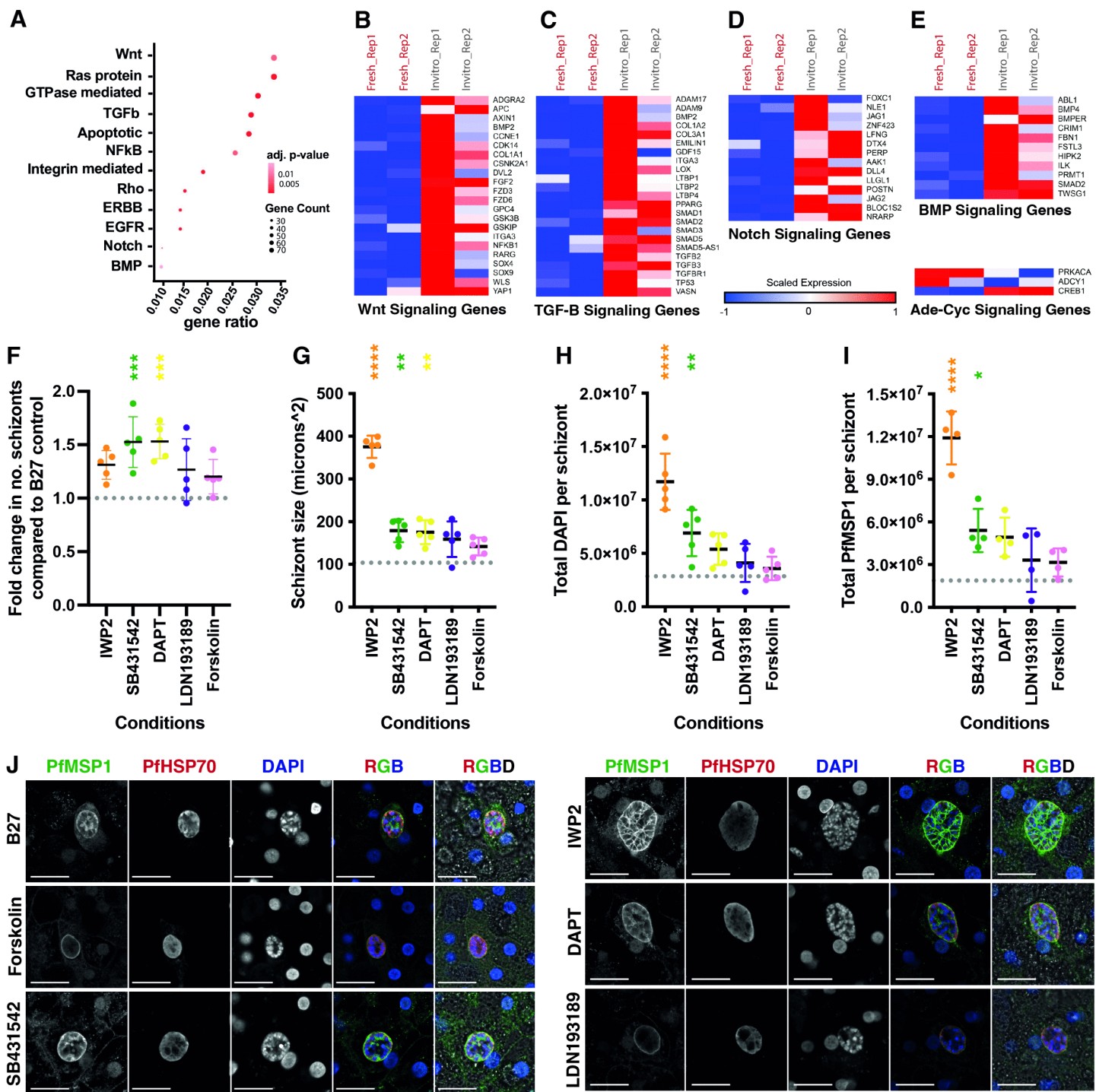

**Fig 2. Effect of signaling pathways on Pf schizont development. A)** Dot-plot depicting the signaling pathways exhibiting higher activity in *in vitro* cultured hepatocytes as compared to freshly isolated hepatocytes. Heatmaps representing normalized expression of genes from **(B)**Wnt, **(C)**TGFβ, **(D)**Notch, **(E)**BMP and Adenylate cyclase signaling pathways deregulated in cultured *vs* freshly isolated hepatocytes. The data is scaled and representative of 2 biological replicates of RNA sequencing. The effect of specific host pathway inhibitors (IWP2, SB431542, DAPT, LDN193189 and Forskolin) on respectively schizont numbers (F; each dot is the average count of two technical replicates), size of schizonts (G; each dot is the median of at least 100 schizonts), total DAPI per schizont (H; each dot is the median of at least 100 schizonts), and total PfMSP1 per schizont (I; each dot is the median of

at least 100 schizonts) on day 5 p.i.. The grey dotted line (per graphs F-I) shows the median measurements of schizonts grown in B27 control (from four independent experiments). The p-values from a Dunnett's multiple comparisons test are shown – see supplementary information for full p values (S1 Data). **J)** Representative confocal images (representative examples from four independent experiments) showing schizonts grown in B27 alone and B27 supplemented with Forskolin, SB431542, IWP2, DAPT and LDN193189 on day 5 p.i. stained with PfMSP1, PfHSP70 antibodies and DAPI. Scale bar is 25 microns.

upregulated in IWP2 treated cells revealed enrichment of metabolism associated terms including fatty acid, alcohol and steroid metabolism (Fig 3I). Among the biological processes suppressed upon IWP2 treatment, we found enrichment of extra-cellular matrix organization, surface-adhesion, epithelium migration and response to TGF-β and Wnt-signaling (Fig 3J). This indicates that IWP2 treatment is able to partially revert the deregulation of key biological pathways observed in PHHs during culture. IWP2 treatment primarily suppressed genes (252 genes) that are otherwise upregulated during *in vitro* culture in B27 medium (as described above, Figs 4K, 1G, and 1H). We, hence, conclude that IWP2 treatment improves the metabolic profile of cultured hepatocytes by suppressing the epithelium to mesenchymal transition, and is associated with an improved developmental kinetics of liver schizonts.

### The activation status of the Wnt pathway determines the size of Pf liver stage schizonts

We subsequently projected the gene-expression changes induced by the IWP2 treatment on elements of the Wnt signaling pathway and as expected identified a modulation of core signaling components (e.g., suppressed expression of ligand: WNT; receptor: Frizzled; transcription factor: TCF/LEF) (Fig 3L). The most likely effect of the repression of the signaling pathway is the downregulation of associated target genes and consequently, an alteration of the cellular state. Therefore, we investigated the activity of key transcription factors (via quantifying the deregulation of their targets genes). As expected, the Wnt-associated transcription factors (e.g., beta-catenin/CTNNB1, TFC7L2) displayed increased activity during *in vitro* culturing and many of their target genes displayed clear down-regulation upon IWP2 treatment (S6B-D Fig). Next to these we also observed altered activity of the SMAD (TGF-β) and AP1 (noncanonical WNT) transcription factors as inferred from deregulation of their target genes, which is likely due to an indirect effect of WNT deregulation (S6B Fig). This agrees with published literature that shows the interlinked nature of the WNT and TGF-β pathways [31–33]. However, given the lack of impact of the TGF-β inhibitor (SB431542) alone on schizont size (Fig 2G), we are convinced of the Wnt pathway's impact on parasite development.

To further examine the impact of the Wnt pathway on Pf schizont size, we treated PHHs with the well-known Wnt activator CHIR99021 prior and during PfNF135 and PfNF175 infection, given that inhibitor of Wnt (IWP2) showed enhanced schizont size and maturation (Fig 4A-4D). PfNF54 in this context was not tested due to lack of sporozoite availability. Compared to the B27 control, the number of schizonts were not altered after CHIR99021 treatment (S4A-B Fig). Schizonts in CHIR99021-treated PHHs were significantly smaller than B27 cultured schizonts on day 5 p.i. for both Pf strains. There was also a significant size decrease for PfNF135 (Fig 4A) but not for PfNF175 (Fig 4B) at day 7 which may be the result of arrested development of PfNF175 under B27 conditions. CHIR99021 treatment also resulted in a relative decrease in PfMSP1 expression significant for PfNF175 on day 7 p.i.. Furthermore, pretreatment of hepatocytes with IWP2 or CHIR99021 was followed by return to WBH after NF175 infection, schizont features (i.e., size, DAPI content and PfMSP1 level) were comparable to schizonts observed in hepatocytes cultured in WBH the entire time. The disappearance of the phenotype upon removal of the treatment (IWP2 or CHIR99021 in B27) indicates the specificity of phenotype to the treatment. Thus, both inhibition and activation of the Wnt pathway in hepatocytes affects predominantly schizont size and maturation. Collectively, this demonstrates the influence of the host Wnt signaling pathway on schizont development for two parasite strains, PfNF135 and Pf175, in PHHs.

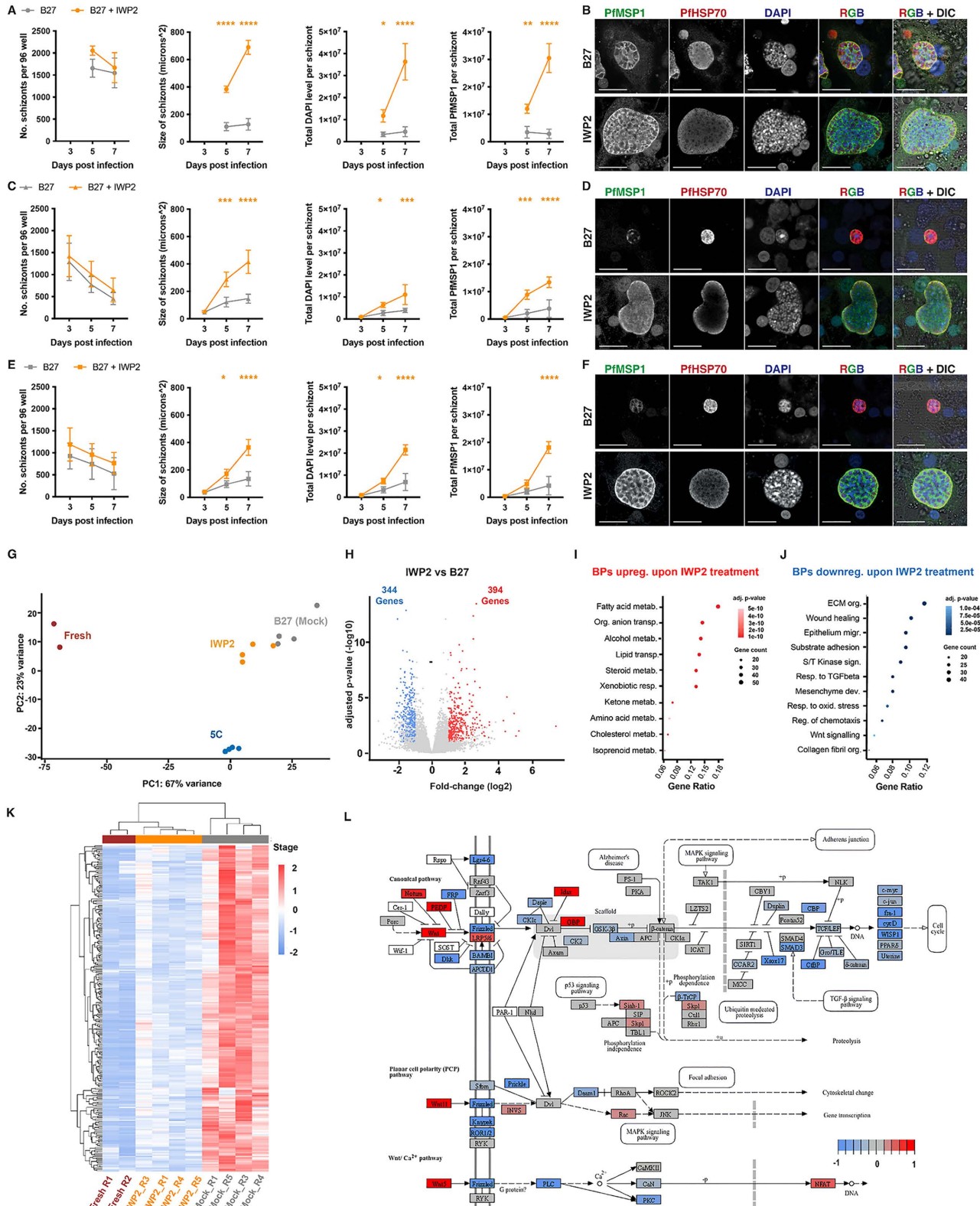

**Fig 3. Effect of IWP2 on developmental kinetics of NF175 (A-B). NF135 (C-D) and NF54 (E-F) schizonts.** For each Pf strain, the number schizonts, the size of schizonts, the total DAPI content and total PfMSP1 content are shown from left to right. The schizonts are either grown in hepatocytes

cultured in either B27 (grey) or B27 supplemented with IWP2 (orange) for 6 days prior to infection and during schizont development on days 3, 5 and 7 p.i. Note briefly that day 3 schizonts of NF175 was not examined. For the number of schizonts, the mean of three independent experiments, each with 2-3 internal replicates is shown. For the schizont size and the total DAPI per schizont, at least 100 schizonts per each of three independent experiments (with two replicates per experiment) were measured. The mean of the median from each experiment is plotted and the graph shows the mean with standard deviation. For the total MSP1 content, at least 100 schizonts per each of three independent experiments were measured. The median is plotted, and the graph shows the mean with standard deviation. A Dunnett's multiple comparisons test is performed: see supplementary information for full p values (S1 Data). **B, D, F)** Representative confocal images (from three independent experiments) showing NF175, NF135 and NF54 schizonts grown in B27 alone and B27 supplemented with IWP2 on day 7 p.i. stained with MSP1, HSP70 and DAPI respectively. Scale bar is 25 microns. **(G)** PCA plot highlighting the unique transcriptional states of freshly isolated ("fresh"), B27 cultured, IWP2 or 5C-treated hepatocytes. **H)** Volcano plot highlighting differential gene expression between IWP2-treatmentand"mock"-cultured hepatocytes. Dot-plots depicting the biological processes (BPs) upregulated (**I**) or downregulated (**J**) in IWP2-treated as compared to "mock" cultured hepatocytes. **K)** Gene expression heatmap highlighting the genes upregulated in mock cultured hepatocytes and suppressed upon IWP2 treatment. The expression counts have been normalized and scaled for plotting. **L)** Schematic overview of genes involved in the Wnt signaling pathway that are modulated by IWP2 treatment. The Fold-change (log2) for the genes are scaled and plotted in blue-red color schematic.

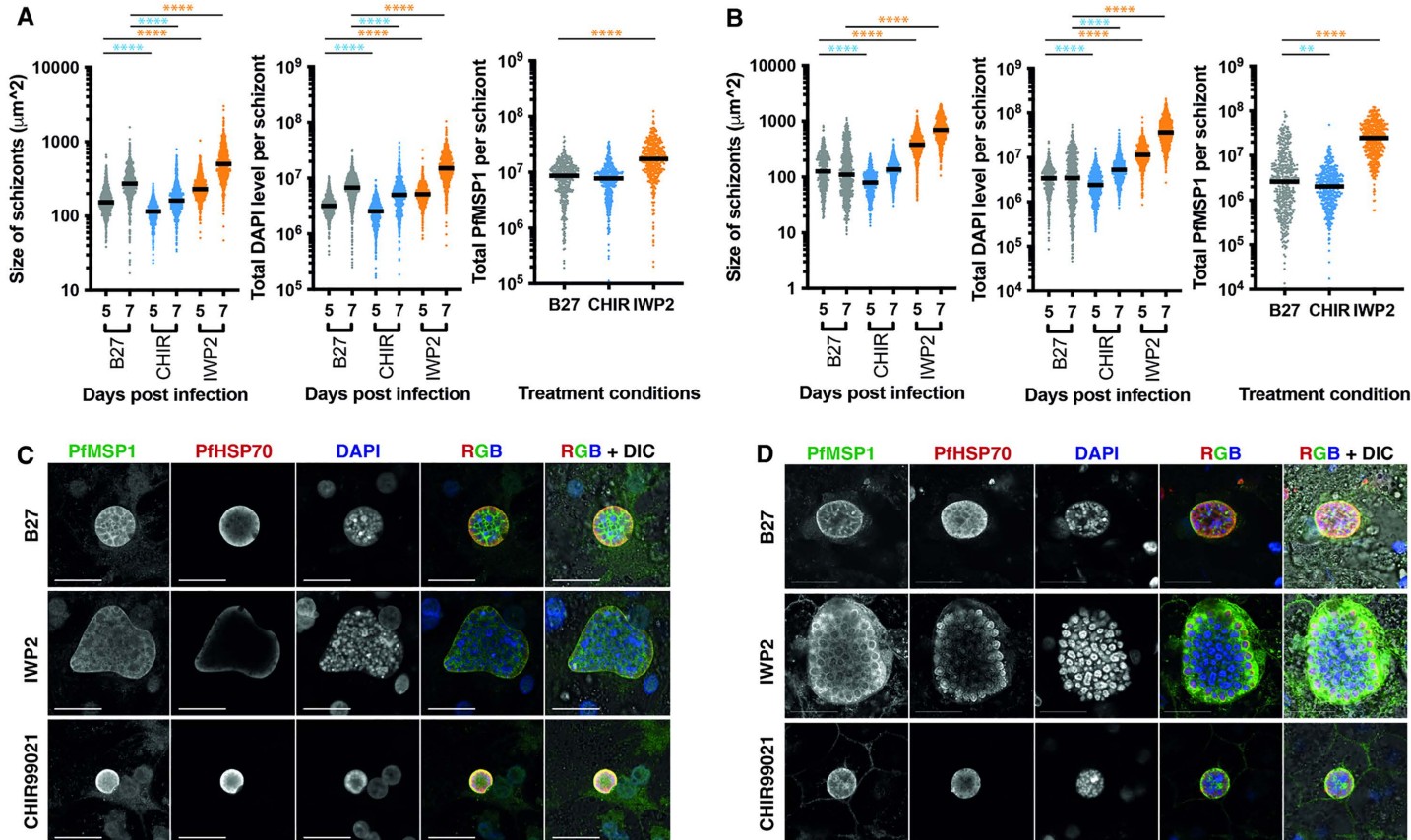

**Fig 4. The impact of the inhibition and activation of the Wnt signaling pathway on Pf schizont size.** Hepatocytes treated with B27 or B27 supplemented with IWP2/CHIR99021 and infected with either NF175 (**A**) or NF135 (**B**). Schizont size (left), and total DAPI (middle) content are measured on days 5 and 7 post infection. The total amount of PfMSP1 per schizont was measured on day 7 p.i. (right). Each dot represents a schizont with at least 100 schizonts measured for three independent experiments. The median is shown, and a Dunn's multiple comparisons test is performed with the p values displayed: see supplementary information for exact p values (S1 Data). Representative confocal images (from three independent experiments) showing NF175 (**C**) and NF135 (**D**) schizonts B27 or B27 supplemented with IWP2/CHIR99021 on day 7 p.i. stained with MSP1, HSP70 and DAPI. Scale bar is 25 microns.

## Discussion

In this study, we show that PHHs in *in vitro* culture quickly lose their ability to sustain a Pf liver stage infection, irrespective of the parasite strain. This loss of permissibility coincides with the up-regulation of specific signaling pathways in PHHs. Inhibition of individual up-regulated pathways (Wnt, TGF-β, and Notch) results in enhanced or maintained hepatocyte permissibility leading to an improvement in schizont development. In addition, we identify the Wnt-pathway as an important host factor regulating cellular features of the host hepatocyte important for Pf liver stage maturation.

The combined repression of four of the up-regulated pathways (Wnt, TGF-β, Notch and BMP) with an adenylate cyclase activator (Forskolin) the 5C cocktail has been shown to greatly benefits hepatitis B virus infection in PHHs [19]. However, such inhibitory cocktail appears to be detrimental for hepatocyte permissiveness and development of Pf as shown in this study (S4 Fig). This could be due to the differing intracellular host requirements by the two distinct pathogens: for example, the intrahepatic growth of HBV is 72 hours compared to the 7 days of Pf. Furthermore, 5C treatment exerted transcriptional and/or cellular changes beyond the prevention of EMT (S6 Fig). In case of Pf, we show that inhibition of specific individual host signaling pathways rather than simultaneous inhibition of all pathways generate strong improvements of parasite development.

The inhibition of the Wnt pathway in cultured hepatocytes has a marked effect on parasite size and maturation. This increase in parasite size is concordant with the upregulation of genes involved in lipid, alcohol and steroid metabolism (Fig 4). While general metabolic health of the hepatocyte is conceivably beneficial for parasite growth, fatty acid synthesis in particular is relevant to parasite development and formation of hepatic merozoites [34]. Furthermore, upon Wnt inhibition we find a downregulation of genes typical for the EMT. In the context of *in vitro* cultures, the EMT is mainly due to the loss of tissue environment but can also occur *in vivo* during liver fibrosis [35] and might influence liver stage infection.

We are unable to compare schizonts grown in IWP2-treated PHHs to those that grown in freshly isolated PHHs for logistical reasons. However, to ascertain the impact Wnt inhibition (via IWP2) on parasite development, we plotted the size of IWP2-treated schizonts against published data [23] which shows the size of the parasites in freshly cultured PHHs (without IWP2). This is comparable to the sizes observed in PHHs cultured for 5–7 days in WBH media (prior and during infection) for all three parasite strains (S7 Fig). This combined with the reduction of schizont size down to WBH levels upon removal of IWP2 from the medium during infection (S5 Fig) are all suggestive of the Wnt pathway having a specific impact on schizont size and maturation.

While the most logical explanation for the effect of IWP2 is via the modulation of the signaling pathway and cellular state of the host hepatocyte, we cannot fully exclude the possibility for a direct effect of IWP2 on parasite development. However, we have no reasons to suspect it to be the case for the following: i) we show that IWP2 has a clear effect on the cellular and transcriptional state of the host cell, which likely influences parasite development; ii) since activation of the Wnt pathway via an agonist leads to the opposite effect on parasite development, it appears more likely that the modulation of the host cell rather than the direct effect of IWP2 would be responsible for the increase growth phenotype; and iii) IWP2 is a specific inhibitor of porcupine, a protein only present in multicellular animals [36].

While our study highlights the relevance of the inhibition of the Wnt signaling pathway in *in vitro* cultures, this observation might also be relevant for *in vivo* zonal differentiation of hepatocytes. It has been well established that within the liver lobule hepatocytes display marked differences in gene expression and metabolism along the portal central axis (also referred to as zonation, [37]) and components of the Wnt pathway are predominantly present in zone 3 hepatocytes and heavily involved in maintaining hepatic zonation [38]. We have previously shown that Pf parasites strongly prefer zone 3 (pericentral) hepatocytes resulting in better parasite development [23], as similarly shown for Pb [39]. While zone 3 hepatocytes possess all components of the Wnt pathway, i.e., the surface receptor and intracellular signaling components, it is possible that external Wnt ligand/signaling proteins may be secreted by hepatocytes from neighbouring zones or other non-hepatocyte cells present in the liver [40] thereby playing a critical role in determining schizont size.

The number of schizonts in a culture can be determined by invasion rate or ability of the invaded parasite to survive and/or develop properly in the infected hepatocytes. To investigate why there is an improved schizont number in IWP2 treated PHHs, we examined the expression of four host receptors (SR-B1, CD81, CD36 and EphA2) that have been implicated to be involved in Pf invasion of hepatocytes across the different culture conditions (S8 Fig). There was a dramatic decline in expression of all receptors in *in vitro* cultured (B27/mock) hepatocytes compared to freshly isolated primary hepatocytes. This likely explains the decline in parasite invasion rates over extended culture duration (Fig 1B). IWP2 treatment showed minor significant improvement on SR-B1 expression and a minor non-significant improvement to CD81 expression while no change or even a decrease in the expressions of CD36 and EphA2 (respectively). Furthermore, there was a significant drop in the expression of AQP3 in the *in vitro* culture (compared to fresh). AQP3 is a host factor co-opted by Pf for supporting its intra-hepatic growth [8]. IWP2 treatment resulted in a minor but non-significant increase in the expression of this host factor.

Inhibition of the TGF-β (SB431542), Notch (DAPT), BMP (LDN193189) and the activation of the adenylate cyclase (Forskolin) pathways all showed improved schizont size compared to B27. This could be attributed to their dynamic interaction with the Wnt signaling pathway. In forskolin treated schizonts, activated adenylate cyclase leads to increased levels of cAMP which in turn can activate the Wnt pathway [41,42]. This can be seen in Fig 2G-2I where Forskolin treated schizonts were smaller, containing less DAPI and less mature compared to the schizonts treated with other compounds investigated. The relationship between the BMP pathway (LDN193189) and the Wnt pathway is less obvious as the BMP pathway can either inhibit or activate the Wnt pathway depending on the presence of p53 or the loss of SMAD4, respectively [43]. Inhibition of the Notch pathway (by DAPT) maintains PHHs in their hepatocyte lineage (as opposed to a more bile ductal phenotype) as well as indirectly inhibiting the Wnt pathway [44]. This could hence explain the significant increase in the size of schizonts grown in DAPT-treated PHHs compared to the control B27 medium. Finally, TGF-β signaling leads to the activation of the Wnt pathway [45] therefore its inhibition results in schizonts with increased size, nuclear content, and maturation.

Despite their differences in infectivity and phenotype, schizonts of all tested Pf strains are significantly larger and more mature in IWP2-treated PHHs than their counterparts in B27-treated PHHs. This larger size becomes significant during the process of schizont growth, i.e., after day 3 p.i. and becomes prominent from day 5 p.i. onwards: inhibition of Wnt signaling between infected- and uninfected cell may allow for better expansion of the growing schizont with less of host restriction. It has been estimated that an infected hepatocyte can expand up to 200 times its original volume to accommodate the growing Pf schizont [46]. This must necessitate communication between the infected and surrounding uninfected hepatocyte and the Wnt signaling pathway may serve here as communication platform [33,35–37]. IWP2 specifically inhibits the enzyme porcupine (Porcn), which is involved in transport of the Wnt ligands [47]. One could hypothesize that neighbouring hepatocytes keep their strict size and volume via the steady basal level of Wnt signaling and thereby limiting the growth of Pf schizonts [48] (S9 Fig).

In conclusion, PHHs cultured in standard medium (WBH) rapidly lose their permissiveness to Pf liver stage development. This loss, however, can be substantially delayed by the usage of a defined serum replacement (B27) on top of a William's E Glutamax base. Furthermore, we identified three specific host signaling pathways (Wnt, TGF-β and Notch) to modulate Pf liver stage development. Inhibition of Wnt signaling controls the size of the developing Pf schizont and is therefore a key factor in determining Pf developmental kinetics for multiple parasite strains. Transcriptome analysis of Wnt inhibitor treated PHHs indicates that improvement of hepatic metabolism, prevention of EMT and cell division lead to the production of more mature liver stage parasites. Together, these findings provide steps forward in our understanding of the intricate interaction between hepatocyte and liver stage parasite, unlocking rational approaches for future therapeutic interventions.

## Materials and methods

### Ethics Statement

Primary human liver cells were freshly isolated from remnants of surgical material. The samples were anonymised and general approval for their use was granted in accordance with the Dutch ethical legislation as described in the Medical Research (Human Subjects) Act. It was confirmed by the Committee on Research involving Human Subjects in the region

 **PLOS** · **Pathogens**

of Arnhem-Nijmegen, the Netherlands. Approval for use of remnant, anonymized surgical material for transcriptome analysis was specifically confirmed by the Committee on Research involving Human Subjects, in the region of Arnhem-Nijmegen, the Netherlands (CMO-2019–5908).

## PHH isolation from liver segments

Primary human hepatocytes were isolated from patients undergoing elective partial hepatectomy as previously described [23,27]. Freshly isolated hepatocytes are suspended in complete William's B medium (WB): William's E with Glutamax (ThermoFisher Scientific: 32551087), 1x Insulin/transferrin/selenium (ThermoFisher Scientific: 41400045), 1mM sodium pyruvate (ThermoFisher Scientific: 11360070), 1x MEM Non-essential amino acid solution (ThermoFisher Scientific: 11140035), 100 units/ml Penicillin-Streptomycin (ThermoFisher Scientific: 15140122) and 1.6µM of dexamethasone (Sigma Aldrich: D4902). PHHs were plated at 62,500 cells per well in 96 well format and kept in a 37°C (5% $CO_2$) incubator with daily medium changes 96-well plates (Falcon: 353219) precoated with Type 1 collagen solution from rat tail (Sigma Aldrich: C3867).

## Generation of sporozoites for liver infection

Pf asexual and sexual stages were cultured in a semi-automatic system as described [49–51]. *Anopheles stephensi* mosquitoes were reared at the Radboud University Medical Center Insectary (Nijmegen, the Netherlands) in accordance with standard operating procedures. Salivary glands from infected mosquitoes (day 14 to day 23 post infection) were hand-dissected and collected in WB medium. Collected glands were homogenized using home-made glass grinders and sporozoites were counted in a Burker-Turk chamber using phase-contrast microscopes. Immediately prior to infection of human hepatocytes, the sporozoites were supplemented with heat-inactivated human sera (HIHS) at 10% of the total volume (i.e., WBH). The multiplicity of infection is 0.8 (e.g., 50,000 sporozoites to 62,500 hepatocytes) or 1 (62,500 sporozoites to 62,500 hepatocytes) depending on the yield of the sporozoite batches. The MOI is the same within an experiment/biological replicate but may differ between experiments (as it depended on the sporozoite yield of the dissected mosquitoes.

## Medium compositions and duration

*Different plating time after isolation (*Fig 1B* and *1C*)*: Isolated PHH were plated in WB medium. The following day, it was changed to WBH: this medium was refreshed daily until the end of the experiment on day 26 post plating. In Fig 1D and 1E, isolated PHH were again plated in WB medium. The following day, it was changed to William's E with Glutamax (ThermoFisher Scientific: 32551087) supplemented with 1x B27 Supplement (ThermoFisher Scientific: 17504044) and 100 units/ml Penicillin-Streptomycin (ThermoFisher Scientific: 15140122) which was referred to as B27 medium or mock in the article. On the day of infection, B27 medium was removed and replaced with sporozoites suspended in WBH for three hours. After the infection process has occurred (i.e., after three hours), the WBH (sporozoite) medium was replaced with B27 medium until the conclusion of the experiment.

*Different medium treatments* Isolated PHH were plated in WB medium. The following day, it was changed to the following medium conditions: B27 or B27 supplemented with 20µM of Forskolin (Enzo Life Sciences: BML-CN100–0010), or 10µM of SB431542 (Tocris: 1614), or 0.5µM of IWP2 (Tocris: 3533), or 5µM of DAPT (Tocris: 2634) or 0.1µM of LDN193189 (Tocris: 6053) or the combination of all the compound inhibitors (i.e., 5C). The hepatocytes were kept on this treatment for another 6 days (i.e., 7 days post plating) and then infected with sporozoites where the medium composition changes to WBH for three hours. After three hours, the monolayers were returned to their respective treatments. Media was refreshed on a daily (24hour) basis.

*Investigating the Wnt pathway* Isolated PHH were plated in WB medium. The following day, it was changed to the following medium conditions: B27 or B27 supplemented with 0.5µM of IWP2 or 3µM CHIR-99021 (Sigma-Aldrich: SML1046).

The hepatocytes were kept on this treatment for another 6 days (i.e., 7 days post plating) and then infected with sporozoites where the medium composition changes to WBH for three hours. After three hours, the monolayers were returned to their respective treatments. Media was refreshed on a daily (24hour) basis.

### Immunofluorescence readout

Monolayers were fixed with 4% paraformaldehyde (ThermoFisher Scientific: 28906) for 10 minutes and permeabilised using 1% Triton for 5 minutes. The samples are stained with the various primary Pf or human antibodies: Rabbit PfHSP70 at 1:75 dilution (StressMarq Biosciences: SPC186), Mouse PfMSP1 at 1:100 dilution (Sanaria and NIH/NIAD: AD233), and mouse human glutamine synthetase at 1:100–250 dilution (Abcam: ab64613). Secondary antibodies were used at these following dilutions: Goat anti-rabbit Alexa Fluor 594 at 1:200 dilution (ThermoFisher Scientific: A11012) and goat anti-mouse Alexa Fluor 488 at 1:200 dilution (ThermoFisher Scientific: A11029). DAPI was used at 300 nM to stain the nuclear material of the monolayer.

### Microscopy

The Zeiss LSM880 with Airyscan at 63x objectives (oil) and 2x zoom were used for detailed images. For high content images, the Zeiss Axio Observer Inverted Microscope Platform with AI assisted experimental startup was used. The images were acquired at 20x objectives with a numerical aperture of 0.8.

### Data analysis using FIJI

*Infection rate* For each well, 77 images were acquired in a tiled format. Approximately half (i.e., 39) images were counted on FIJI [52] for NF135 and NF175 infections. All the tiles were counted for NF54 due to the lower infection rate. The number of hepatic nuclei were counted for 1% of the total image (i.e., 7–10 images) and then extrapolated to get number of PHHs per well.

*Measurement of schizont size* See Yang et al for further details [23]. Images obtained on the high content microscope were opened in FIJI. Random images were chosen until at least 100 schizonts were measured (per well) unless the infection is with NF54 (at least 50 schizonts). Schizonts were selected using the region of interest (ROI) tool based on PfHSP70 positivity (red channel) and measured. This ROI mask is applied onto the other colour channels, i.e., blue and green to obtain values of nuclear content (DAPI) and hGS or PfMSP1 signal. For hGS and PfMSP1, background non-specific staining was considered and subtracted from the final signal. Further details regarding the methods can be found in [21,23].

### RNA Isolation and RNA sequencing library preparation

Freshly isolated or in vitro cultured hepatocytes (mock or treatment) were homogenized in TRIzol solution and cryopreserved at -80°C. RNA was subsequently isolated using the Zymo research Direct zol RNA purification kit (Cat. No. #R2053). The isolated RNA was assessed for quality and quantified using Nanodrop and agarose gel electrophoresis. RNA sequencing libraries from isolated/purified RNA were prepared using the Kapa mRNA Hyperprep Kit (Cat. No. #KK8581) using slight modifications of the manufacturer's instructions for mRNA enrichment method. The input total RNA for the various libraries was selected between 150ng, 250ng or 500ng. The enriched mRNA was fragmented at 94°C for 6min and dA tailing performed at 55°C for 15min. In order to better capture the AT rich *Pf* transcriptome, we modified the PCR protocol as follows: Initial denaturation at 98°C for 2min; Cycling denaturation at 98°C for 20sec, annealing + extension at 62°C for 2min; final extension at 62°C for 3min. We selected PCR cycles for library amplification (12, 11 and 10 respectively) based on starting input total RNA amount. The amplified libraries were quantified using Denovix dsDNA high sensitivity kit (Cat. No. #TN145)on a Qubit fluorometer. Further qualitative assessment of library size distribution was

performed using the Agilent high sensitivity DNA kit (Cat. No. #5067–4626) on the Agilent 2100 bioanalyzer platform. Each library was sequenced in 42 bp paired-end format for roughly 18 million reads on the Illumina Nextseq 500 platform.

## RNA sequencing data analysis

The sequencing endline fastq files were quality checked using FASTQC (ver. 0.11.9). The reads were subsequently trimmed for quality (-q 30) and adapter removal using the Trim_Galore software (ver. 0.6.7). The trimmed reads were aligned onto the *Homo sapiens* ver. 38 genome using STAR aligner (ver. 2.7.10a). The reads mapped onto the respective genomes were counted using the –quantMode GeneCounts option in STAR. Reads mapping only to sense strand were corrected for batch effects stemming from different hepatocyte donors using the Combat-seq tool (sva package ver. 3.44.0). The batch corrected counts were used for subsequent differential expression analysis (for conditions/treatments) in DESEq2 package (ver. 1.36.0) in RStudio (ver. 4.2.2 "Innocent and Trusting"). PCA plot for the various sequenced libraries/samples was generated using plotPCA command. Volcano plots for differentially expressed genes and genes of interest were generated using custom scripts and ggplot2 (ver. 3.4.2) in R. Gene Ontology enrichment analysis was performed in using the clusterProfiler tool (ver. 4.4.4) with subsequent dot-plots generated using inbuilt functions in ggplot2. Heatmaps for differentially expressed genes were generated using the Morpheus online tool by Broad Institute and custom scripts in R (ggplot2). KEGG pathway schematic for Wnt signaling was generated using the pathview tool package (ver. 1.46.0) in R. Transcription factor activity analysis was performed using the decoupleR package (ver. 2.6.0 utilizing collecTRI database) from the Julio Saez-Rodriguez lab [53,54]. Bar-plot visualization for TF activity scores was done in GraphPad Prism 8.

## Statistical analysis

For the majority of the experiments, at least three biological replicates were performed with two technical replicates. All statistical tests were performed using Prism 10. See fig legends for details regarding the statistical tests.

## Supporting information

**S1 Fig. The impact of heat inactivated human sera on Pf schizont development.** PHH monolayers were cultured in either William's B (WB) or William's B supplemented with 10% heat inactivated sera (WBH) for six days p.p. and infected with either NF175 (A) or NF135 (B) or NF54 (C). Data are shown for respectively schizont number, size, total DAPI content, percentage of human glutamine synthetase (hGS) positive schizonts, total hGS level per schizont and total merozoite surface protein 1 (MSP1) level per schizont. As for the number of schizonts, each dot represents an independent experiment showing the mean of two replicates. As for the size and total DAPI content graphs, each dot represents an independent experiment showing the mean of the median of at least 100 schizonts per replicate (2 replicates) were measured (except in NF54 condition where there is not enough schizonts present). For the percentage of hGS positive schizonts, each dot represents an independent experiment where at least 100 schizonts were examined from one replicate. For the total hGS level and total MSP1 level, each dot represents an independent experiment of the median from at least 100 schizonts (except in NF54 condition where there is not enough schizonts present). The error bars show the mean with the standard deviation. The p-values are generated by performing an unpaired t-test between WBH and WB and can be found in S1 Data (*** = 0.0001).
(TIF)

**S2 Fig. Effect of specific inhibitors of PHH gene-signaling pathways on the expression Pf schizont markers A) Schematic experimental setup of Figs 3 and S3.** PHHs were treated for 6 days p.p with either B27 or B27 supplemented IWP2/SB431542/Forskolin/DAPT/LDN193189. The experiment was analysed at day 5 p.i. with NF175 B) Total number of hepatocytes per well with each dot representing the average three wells of one biological replicate. C)Total

hGS per schizont with each dot representing the median of at least 100 schizonts. The percentage of hGS (D) and MSP1 (E) positive schizonts: each dot is an independent experiment where at least 100 schizont is measured. The grey dotted line (per graphs C-E) shows the median measurements of schizonts grown in B27 (from four independent experiments) as each host pathway inhibitor treatment was made in media containing B27. The mean and standard deviation is shown for each graph. The p-values from a Dunnett's multiple comparisons test are displayed. F) Representative confocal images (from four independent biological experiments) showing schizonts grown in B27 alone and B27 supplemented with IWP2, SB431542, DAPT, LDN193189, Forskolin stained with hGS, HSP70 and DAPI. Scale bar is 25 microns.
(TIF)

**S3 Fig. The effect of B27 or B27 supplemented with 5C on *Pf*-permissiveness and schizont development in PHH.** The number of schizonts (A) and hepatocytes (B) on day 5 p.i. for NF135 and NF54 (n = 1) at indicated time points p.p. when cultured in either B27 medium (grey) or B27 supplemented with 5C (blue). Each dot shows the mean of three replicates and the error bars show the standard deviation. C) Experimental setup of graphs D-L. PHHs were cultured for 6 days p.p. with either B27 or B27 with 5C and infected with NF175 parasites and analysed at day 5 p.i. D) Total number of hepatocytes per well with each dot representing the average three wells of one biological replicate. E)The number of schizonts per well was normalised to the number of schizonts counted in WBH (red dotted line). Each dot is the mean of an independent experiment with two replicates. For the schizont size (F) and total DAPI content per schizont (G), each dot represents the mean of an independent experiment of at least 100 schizonts. For the total hGS (H) and MSP1 (I) per schizont, each dot represents the median of an independent experiment of at least 100 schizont. For the percentage of hGS (J) and MSP1 (K) positive cells, each dot represents an independent experiment of at least 100 schizonts. D-K shows the mean with the standard deviation). Representative confocal images (from four independent experiments) showing schizonts grown in WBH or B27 alone or B27 supplemented with 5C on day 5 p.i. stained with hGS (L) or MSP1 (M), HSP70 and DAPI. Scale bar is 25 microns.
(TIF)

**S4 Fig. The impact of the Wnt signaling pathway on numbers of developing schizonts.** A) The number of schizonts in NF135 infected PHHs treated with B27 or B27 supplemented with IWP2/CHIR99021. Each graph shows an independent experiment, each with duplicates per condition. B) Number of schizonts (left) and number of hepatocytes (right) in NF175-infected PHHs treated with B27 or B27 supplemented with IWP2/CHIR99021. Each graph shows the mean and the standard deviation of three independent experiments: each experiments have duplicate wells. There was no statistic difference between conditions within the same day post infection. Representative confocal images (from three independent biological experiments) showing NF135 (C) and NF175 (D) schizonts B27 or B27 supplemented with IWP2/CHIR99021 on day 5 p.i. stained with hGS, HSP70 and DAPI. Scale bar is 25 microns.
(TIF)

**S5 Fig. The impact of Wnt signaling on determining NF175 development in the liver A) schematic showing the media treatment timing and conditions.** Hepatocytes are treated with B27 or B27 supplemented with IWP2/CHIR99021. After exposure to NF175, cultures were either kept in B27 or B27 supplemented with IWP2/CHIR99021 (closed circles) or returned to WBH (IWP2* and CHIR99021* - open circles). The number of schizonts (B), size of schizonts (C), total nuclear content per schizont (D) and total MSP1 per schizont (E) on day 5 post infection. The number of schizonts (F), size of schizonts (G), total nuclear content per schizont (H) and total MSP1 per schizont (I) on day 7 post infection. Each dot represents an independent biological donor where the median of at least 100 schizonts were measured. Dunnett's multiple comparison test (to WBH) were performed for each graph, and the statistical significance is indicated: see supplemental information excel sheet for the full p values (S1 Data).
(TIF)

**S6 Fig. IWP2 treatment modulates expression of target genes downstream of Wnt responsive transcription factors.** A) PCA plot of all replicates of freshly-isolated, "mock-cultured", IWP2 or 5C-treated hepatocytes. B) Bar-plot depicting the scaled activity scores of key Wnt, TGFβ and non-canonical Wnt transcription factors across freshly-isolated hepatocytes, "mock-" and IWP2-treated hepatocytes. C-D) Volcano plots depicting a repressive effect of IWP2 treatment on the target genes underlying key Wnt signaling transcription factors CTNNB1 (β-catenin; C) and TCF7L2 (D). Genes that are positively modulated by these TFs and suppressed upon IWP2 treatment are highlighted in blue.
(TIF)

**S7 Fig. Comparison of PfNF175 (A), PfNF54 (B) and PfNF135 (C) schizont growth in PHHs cultured in WBH, B27 or B27-IWP2 for 6–8 days prior to infection to growth in freshly isolated PHHs cultured in WBH (historical data).**
(TIFF)

**S8 Fig. Expression of key host factors needed for Pf schizont invasion and/or development in freshly isolated hepatocytes, and cultured hepatocytes in B27 (mock), B27-IWP2, or B27-5C medium.**
(TIF)

**S9 Fig. Schematic of a possible mechanism in which the Wnt pathway controls Pf schizont development. A)**Under untreated and uninfected condition, there is a steady level of beta-catenin (β-catenin). In addition to its role in gene transcription, β-catenin is the intracellular component that links the cadherin adhesion molecules to actin filaments and therefore controls the "rigidity" of the cells in relation to its neighbours. **B)** In an infected hepatocyte (after 3 days post infection), the growing Pf schizont takes up so much of the hepatocyte volume that the trafficking of the Wnt ligands to the surface is disrupted. Wnt receptor of the neighbouring uninfected cells are not activated, and β-catenin molecules are degraded due to the phosphorylation of the enzyme, glycogen synthase kinase 3 (GSK3): this allows some flexibility in the uninfected neighbouring cells to accommodate the growth of the infected cell. However, in these (uninfected cells), Wnt ligands are still present and can interact with the existing Wnt receptors on the infected hepatocytes to maintain some β-catenin in the infected cell, thus limiting the growth/size of the schizont. **C)** Under IWP2 treatment, Wnt ligands are not trafficked to the surface of both uninfected and infected hepatocytes due to the inhibition of the enzyme porcupine (target of IWP2). Porcupine "labels" (via palmitoylation) Wnt ligands for correct trafficking to the plasma membrane. As a result, Wnt receptors on both uninfected and infected cells are not activated and existing β-catenin are degraded leading to reduced connection between cadherin and actin filaments (i.e., cell-cell contacts) and ultimately reducing rigidity. **D)** Under CHIR99021 (GSK3 inhibitor) treatment, β-catenin in both uninfected and infected cells cannot be phosphorylated nor degraded. Both cell types are more rigid due to the improve connection between actin filaments and cadherin which severely limits the size of the Pf schizont.
(TIF)

**S1 Data. Each tab of the excel sheet presents a table of the statistical tests performed per figure with the associated P-values.**
(XLSX)

## Acknowledgments

We are grateful for R. Stoter, W. Graumans, M. Vegte-Bolmer, A. Pouwelsen, L. Pelser-Posthumus and J. Kuhnen of the Malaria and Insectary Unit at the Radboud University Medical Center for parasite, mosquito and sporozoite production. We would like to thank the Microscopic Imaging Center (MIC) of the Radboud University for access to its facilities.

## Author contributions

**Conceptualization:** Haoyu Wu, Robert W. Sauerwein, Richárd Bártfai, Annie S. P. Yang.

**Data curation:** Abhishek Kanyal, Geert-Jan van Gemert, Haoyu Wu, Alex van der Starre, Johannes H.W. de Wilt, Annie S. P. Yang.

**Formal analysis:** Abhishek Kanyal, Annie S. P. Yang.

**Funding acquisition:** Richárd Bártfai, Annie S. P. Yang.

**Investigation:** Alex van der Starre.

**Methodology:** Alex van der Starre.

**Writing – original draft:** Robert W. Sauerwein, Richárd Bártfai, Annie S. P. Yang.

**Writing – review & editing:** Abhishek Kanyal, Geert-Jan van Gemert, Haoyu Wu, Johannes H.W. de Wilt, Teun Bousema, Robert W. Sauerwein, Richárd Bártfai, Annie S. P. Yang.

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
