## [Editor Report · Decision Letter 0]

10 Feb 2025

Modulation of host signalling pathways reveal a major role for Wnt signalling in the maturation of Plasmodium falciparum liver schizonts

PLOS Pathogens

Dear Dr. Yang,

Thank you for submitting your manuscript to PLOS Pathogens. After careful consideration, we feel that it has merit but does not fully meet PLOS Pathogens's publication criteria as it currently stands. Therefore, we invite you to submit a revised version of the manuscript that addresses the points raised during the review process.

Please submit your revised manuscript within 60 days Apr 10 2025 11:59PM. If you will need more time than this to complete your revisions, please reply to this message or contact the journal office at plospathogens@plos.org. Please include the following items when submitting your revised manuscript:

We look forward to receiving your revised manuscript.

Kind regards,

Dominique Soldati-Favre

Section Editor

PLOS Pathogens

Dominique Soldati-Favre

Section Editor

PLOS Pathogens

Editor-in-Chief

PLOS Pathogens

orcid.org/0000-0003-2946-9497

Editor-in-Chief

PLOS Pathogens

orcid.org/0000-0002-7699-2064

**Additional Editor Comments :**

Your manuscript was assessed at Review Commons, and the plan for revision addressing the identified shortcomings seems sound. We are open to considering it for publication in PloS Pathogens as long as these issues are thoroughly addressed in the major revision.

**Journal Requirements:**

https://journals.plos.org/plospathogens/s/submission-guidelines#loc-parts-of-a-submission

- TM on page: 10.

5) We have noticed that you referred to Supplementary Figures (S1-S4) in your manuscript. However, there are no corresponding files uploaded to the submission. Please upload them as separate files with the item type 'Supporting Information'.

6) Please add a full list of legends for your Supporting Information files after the references list.

7) Some material included in your submission may be copyrighted. According to PLOSu2019s copyright policy, authors who use figures or other material (e.g., graphics, clipart, maps) from another author or copyright holder must demonstrate or obtain permission to publish this material under the Creative Commons Attribution 4.0 International (CC BY 4.0) License used by PLOS journals. Please closely review the details of PLOSu2019s copyright requirements here: PLOS Licenses and Copyright. If you need to request permissions from a copyright holder, you may use PLOS's Copyright Content Permission form.

Potential Copyright Issues:

i) Figure 5. Please confirm whether you drew the images / clip-art within the figure panels by hand. If you did not draw the images, please provide (a) a link to the source of the images or icons and their license / terms of use; or (b) written permission from the copyright holder to publish the images or icons under our CC BY 4.0 license. Alternatively, you may replace the images with open source alternatives. See these open source resources you may use to replace images / clip-art:

8) Please update your Data Availability statement in the online submission form and ensure that it matches the one provided in the manuscript.

9) Please amend your detailed Financial Disclosure statement. This is published with the article. It must therefore be completed in full sentences and contain the exact wording you wish to be published.

2) If any authors received a salary from any of your funders, please state which authors and which funders..

**Figure resubmission:**

**Reproducibility:**



---

## [Decision Letter · Decision Letter 1]

17 Jun 2025

Modulation of host signaling pathways reveal a major role for Wnt signaling in the maturation of Plasmodium falciparum liver schizonts

PLOS Pathogens

Dear Dr. Yang,

Thank you for submitting your manuscript to PLOS Pathogens. After careful consideration, we feel that it has merit but does not fully meet PLOS Pathogens's publication criteria as it currently stands. Therefore, we invite you to submit a revised version of the manuscript that addresses the points raised during the review process.

Please submit your revised manuscript within 60 days Aug 16 2025 11:59PM. If you will need more time than this to complete your revisions, please reply to this message or contact the journal office at plospathogens@plos.org. Please include the following items when submitting your revised manuscript:

We look forward to receiving your revised manuscript.

Dominique Soldati-Favre

Section Editor

PLOS Pathogens

Editor-in-Chief

PLOS Pathogens

orcid.org/0000-0003-2946-9497

Editor-in-Chief

PLOS Pathogens

orcid.org/0000-0002-7699-2064

**Additional Editor Comments (if provided):**

The reviewers are recognizing the strengths of this manuscript as a technical report. It presents valuable methodological insights such as those investigating *Plasmodium falciparum* liver-stage biology that will likely benefit the field. However, the current version does not convincingly demonstrate a direct involvement of the Wnt pathway in the parasite itself. As such, the broader biological significance of the findings remains uncertain. I therefore recommend a substantial revision, including repositioning the manuscript as a technical report. This should involve modifying the title accordingly and moderating the current claims to better reflect the presented data. In addition, Reviewer 3 has provided useful editorial suggestions that would help clarify the manuscript. These should also be carefully addressed in the revised version.

**Journal Requirements:**

At this stage, the following Authors/Authors require contributions: Alex van der Starre. Please ensure that the full contributions of each author are acknowledged in the "Add/Edit/Remove Authors" section of our submission form.

2) Please amend your detailed Financial Disclosure statement. This is published with the article. It must therefore be completed in full sentences and contain the exact wording you wish to be published.

2) If any authors received a salary from any of your funders, please state which authors and which funders..

3) Thank you for stating 'We have now provided a reviewers token to access the RNAseq data which will be made fully public upon acceptance/publication. To review GEO accession GSE263643:Go to https://www.ncbi.nlm.nih.gov/geo/query/acc.cgi?acc=GSE263643Enter token ehkrsqmwtdofdmv into the box

' Please note that, though access restrictions are acceptable now, your entire minimal dataset will need to be made freely accessible if your manuscript is accepted for publication. This policy applies to all data except where public deposition would breach compliance with the protocol approved by your research ethics board. If you are unable to adhere to our open data policy, please kindly revise your statement to explain your reasoning and we will seek the editor's input on an exemption.".

4)  Please modify figure 5 description on the online submission form..

**Reviewers' Comments:**

Reviewer's Responses to Questions

**Part I - Summary**

Reviewer #1: This paper addresses an important problem - the low efficacy of hepatocyte infection by Plasmodium falciparum and finds a way to improve it. Along this way the authors find two signaling pathways that are playing a role in improved infection. The problem is that these pathways are activated in vivo anyway and thus the paper gives no real insights into a biological problem.

Reviewer #2: This is a revised version of a manuscript where the authors explore the mechanisms by which primary human hepatocytes rapidly lose their ability to support P. falciparum (Pf) liver stage development in culture conditions.

The authors used transcriptomics and identified several pathways potentially involved in the loss of function of PHH cultures, including Wnt. They demonstrate that the Wnt inhibitor IWP2 results in much improved schizont growth and maturation in vitro. In contrast, a Wnt activator has only a modest effect. Importantly, the effect of IWP2 is confirmed using different hepatocyte donors and with three different Pf strains. The manuscript is clearly written and the work is well performed.

The authors addressed in part the comments of the previous reviewers. The authors are to be commended for this extensive work that tackles a major roadblock in malaria research. The work will be extremely useful for researchers working on Pf liver stage biology. However, the biological impact of the data, beyond technical improvement of the culture conditions, is much less clear to me.

Reviewer #3: I have made numerous and extensive edits to the PDF I was sent to review and the editors and authors are welcome to read them and I feel the authors should address them to improve the paper.

Ultimately, I believe that the study is fascinating but lacks scientific rigor and the results section is poorly written which made it challenging for me to fully digest the importance of the work.

**Part II – Major Issues: Key Experiments Required for Acceptance**

Reviewer #1: please change the title, this is misleading as you don't provide a role for wnt signaling in vivo but in vitro. I think the paper is a great and helpful contribution but please don't make it sounds like it's more than it is. Adapt the abstract accordingly, ie tone it down somewhat

Reviewer #2: In culture, PHH rapidly undergo epithelial-mesenchymal transition (EMT) associated with loss of hepatic functions, resulting in poor Pf susceptibility. The main claim of the paper is that Pf parasites depend on Wnt pathway regulation for efficient liver stage development. The claim is based on the observation that pharmacological inhibition of Wnt results in highly improved liver stage development. This effect is clearly documented throughout the manuscript. However, it is unclear whether this is a direct effect of the Wnt pathway on parasite development or merely the result of preserved hepatic functions due to inhibition of EMT. Some elements are missing in the demonstration to help the reader distinguish between direct or indirect effects. In particular, in Fig 1, a comparison of the size of the schizonts when infection is performed in fresh (day 2) versus plated (Day 5-7) PHH in the different conditions (with or without IWP2) would be helpful to distinguish whether IWP2 preserves or promotes liver stage development.

A stronger demonstration of the role of the Wnt pathway during infection would be to use a model where hepatocytes do not dedifferentiate, such as humanized mouse models. In this regard, how do day 5 schizonts in IWP2-treated PHH cultures compare in terms of size with schizonts in liver-humanized mice ? Another way to look at this would be to test the effect of CHIR99021 on freshly isolated PHH.

In the absence of in-depth characterization of the Wnt pathway I agree with Reviewer 1 that the manuscript should rather be presented as a technical report investigating the mean to improve the Pf liver stage culture system.

Fig 1B shows a drop in the number of schizonts when the age of PHH culture increases, pointing at a defect in invasion and/or survival of the parasite. While IWP2 does not seem to overtly affect the number of schizonts, this is an important feature of the PHH culture ageing. B27 medium clearly preserves the number of schizonts, at least for 1 week. Did the authors check for expression of entry/survival factors ? There are clearly two distinct features of PHH that seem to be regulated differently (at least in part), i.e. susceptibility to infection versus ability to promote parasite maturation, and this should be at least discussed in the manuscript.

Is it possible that IWP2 has a direct effect on the parasite? Stopping treatment when sporozoites are added to the culture results in the drop of the effect (Fig S5, IWP2* condition). What happens in the reciprocal condition (addition of IWP2 only with the parasite)? Is there an improvement of parasite maturation?

Reviewer #3: I have made numerous and extensive edits to the PDF I was sent to review and the editors and authors are welcome to read them and I feel the authors should address them to improve the paper.

In one of the key experiments, outlined in Figure 4 and highlighted in my edited PDF, media composition was changed and the reason was not explained. I believe this experiment should be repeated. No discussion is given as to why three strains of Pf were used in the study.

**Part III – Minor Issues: Editorial and Data Presentation Modifications**

Reviewer #1: small errors should be corrected such as

115: delete 'into'

146: rephrase 'neither / nor'

156: add 'signaling' after wnt and tgf-b

159: add 'pathway' after wnt

Reviewer #2: The authors use either “permissiveness” or “permissibility”. This could be homogenized.

Line 115: to into

Line 131. It would be logical in the reading flow to include parasite maturation (size) in Fig 1.

Line 150-157. The text should refer clearly to numbers versus size of the schizonts.

Line 174-175. Lower scores define higher similarity between samples. Then please check the numbers (IWP2: 168; B27: 147).

The 5C cocktail seems to prevent EMT, yet has not such a positive effect on Pf infection in vitro. This raises the possibility that one of the regulated pathways could be required for parasite maturation. Did the authors try different combinations of the inhibitors to search for the most suitable Pf-adapted cocktail?

Discussion line 257-263. The observation that Pf prefers zone 3 hepatocytes, where Wnt signaling is more prominent, seems to contradict the in vitro data and the claim of the manuscript. Could the authors clarify this discrepancy?

Reviewer #3: I have made numerous and extensive edits to the PDF I was sent to review and the editors and authors are welcome to read them and I feel the authors should address them to improve the paper.

PLOS authors have the option to publish the peer review history of their article (what does this mean? ). If published, this will include your full peer review and any attached files.

**Do you want your identity to be public for this peer review?** For information about this choice, including consent withdrawal, please see our Privacy Policy .

Reviewer #1: No

Reviewer #2: No

Reviewer #3: No

**Figure resubmission:**

**Reproducibility:**



---

## [Decision Letter · Decision Letter 2]

10 Oct 2025

PPATHOGENS-D-25-00264R2

Inhibition of Wnt signaling in primary human hepatocytes promotes Plasmodium falciparum liver stage development

PLOS Pathogens

Dear Dr. Yang,

Thank you for submitting your manuscript to PLOS Pathogens. After careful consideration, we feel that it has merit but does not fully meet PLOS Pathogens's publication criteria as it currently stands. Therefore, we invite you to submit a revised version of the manuscript that addresses the points raised during the review process.

Please submit your revised manuscript within 30 days Dec 09 2025 11:59PM. If you will need more time than this to complete your revisions, please reply to this message or contact the journal office at plospathogens@plos.org. Please include the following items when submitting your revised manuscript:

We look forward to receiving your revised manuscript.

Kind regards,

Dominique Soldati-Favre

Section Editor

PLOS Pathogens

Sumita Bhaduri-McIntosh

Editor-in-Chief

PLOS Pathogens

orcid.org/0000-0003-2946-9497

Michael Malim

Editor-in-Chief

PLOS Pathogens

orcid.org/0000-0002-7699-2064

**Additional Editor Comments (if provided):**

While the overall outcome remains promising, the reviewers noted that some of their previous recommendations were not adequately reflected in the revision. One reviewer has provided additional detailed comments and suggestions directly on the PDF to guide the necessary improvements.

Please revise your manuscript carefully, ensuring that all outstanding points are addressed and that the text reads clearly and coherently throughout.

Kindly note that this is a **final opportunity** to revise the manuscript. No further rounds of revision will be permitted after this stage.

@font-face{font-family:"Cambria Math";panose-1:2 4 5 3 5 4 6 3 2 4;mso-font-charset:0;mso-generic-font-family:roman;mso-font-pitch:variable;mso-font-signature:-536870145 1107305727 0 0 415 0;}@font-face{font-family:Aptos;panose-1:2 11 0 4 2 2 2 2 2 4;mso-font-charset:0;mso-generic-font-family:swiss;mso-font-pitch:variable;mso-font-signature:536871559 3 0 0 415 0;}p.MsoNormal, li.MsoNormal, div.MsoNormal{mso-style-unhide:no;mso-style-qformat:yes;mso-style-parent:"";margin:0in;mso-pagination:widow-orphan;font-size:12.0pt;font-family:"Aptos",sans-serif;mso-ascii-font-family:Aptos;mso-ascii-theme-font:minor-latin;mso-fareast-font-family:Aptos;mso-fareast-theme-font:minor-latin;mso-hansi-font-family:Aptos;mso-hansi-theme-font:minor-latin;mso-bidi-font-family:"Times New Roman";mso-bidi-theme-font:minor-bidi;mso-font-kerning:1.0pt;mso-ligatures:standardcontextual;}p{mso-style-noshow:yes;mso-style-priority:99;mso-margin-top-alt:auto;margin-right:0in;mso-margin-bottom-alt:auto;margin-left:0in;mso-pagination:widow-orphan;font-size:12.0pt;font-family:"Times New Roman",serif;mso-fareast-font-family:"Times New Roman";}.MsoChpDefault{mso-style-type:export-only;mso-default-props:yes;font-family:"Aptos",sans-serif;mso-ascii-font-family:Aptos;mso-ascii-theme-font:minor-latin;mso-fareast-font-family:Aptos;mso-fareast-theme-font:minor-latin;mso-hansi-font-family:Aptos;mso-hansi-theme-font:minor-latin;mso-bidi-font-family:"Times New Roman";mso-bidi-theme-font:minor-bidi;}div.WordSection1{page:WordSection

**Journal Requirements:**

**Reviewers' Comments:**

Reviewer's Responses to Questions

**Part I - Summary**

Reviewer #2: (No Response)

Reviewer #3: This is the second revision of a paper that describes how inhibiting wnt signaling in primary human hepatocyte cultures can improve Plasmodium falciparum liver stage development. The primary outcome of the inhibition is clear and unequivocal. However, the paper is still poorly written and changes that reviewers have asked for on two separate occasions have still not been addressed. In many instances the writing lacks clarity and is opaque. The experimentation is sound but the manuscript needs work.

**Part II – Major Issues: Key Experiments Required for Acceptance**

Reviewer #2: The authors have toned down their statements to reflect more accurately the data and changed the title accordingly.

They bring interesting information in their rebuttal, which unfortunately was not included in the manuscript.

In particular, the observation that IWP2 increases schizont size when compared to (historical) fresh hepatocytes is important as it suggests a direct impact of Wnt inhibition on parasite development.

Also, I still believe that the discussion should include a paragraph to discuss parasite numbers (invasion) versus size (development), perhaps showing the entry factor expression data shown in the rebuttal as supplementary data. This is important as well as it shows that the invasion rate might still be a limiting factor when using aged hepatocyte cultures, even with Wnt inhibitors.

Reviewer #3: I don't feel that major modifications to the experimentation is necessary.

**Part III – Minor Issues: Editorial and Data Presentation Modifications**

Reviewer #2: (No Response)

Reviewer #3: There are numerous editorial issues that need to be addressed. I have documented these in my edited PDF.

PLOS authors have the option to publish the peer review history of their article (what does this mean? ). If published, this will include your full peer review and any attached files.

**Do you want your identity to be public for this peer review?** For information about this choice, including consent withdrawal, please see our Privacy Policy .

Reviewer #2: No

Reviewer #3: No

**Figure resubmission:**
---

## [Editor Report · Decision Letter 3]

10 Dec 2025

Dear Dr. Yang,

We are pleased to inform you that your manuscript 'Inhibition of Wnt signaling in primary human hepatocytes promotes Plasmodium falciparum liver stage development' has been provisionally accepted for publication in PLOS Pathogens.

Best regards,

Dominique Soldati-Favre

Section Editor

PLOS Pathogens

Dominique Soldati-Favre

Section Editor

PLOS Pathogens

Sumita Bhaduri-McIntosh

Editor-in-Chief

PLOS Pathogens

orcid.org/0000-0003-2946-9497

Michael Malim

Editor-in-Chief

PLOS Pathogens

orcid.org/0000-0002-7699-2064
---

## [Editor Report · Acceptance letter]

Dear Dr. Yang,

We are delighted to inform you that your manuscript, "Inhibition of Wnt signaling in primary human hepatocytes promotes Plasmodium falciparum liver stage development," has been formally accepted for publication in PLOS Pathogens.

Best regards,

Sumita Bhaduri-McIntosh

Editor-in-Chief

PLOS Pathogens

orcid.org/0000-0003-2946-9497

Michael Malim

Editor-in-Chief

PLOS Pathogens

orcid.org/0000-0002-7699-2064